# Actinomycetes isolated from rhizosphere of wild Coffea arabica L. showed strong biocontrol activities against coffee wilt disease

Mimi Nuguse, Tekalign Kejela *

Department of Biology, College of Natural and Computational Sciences, Mattu University, Mettu, Oromia, Ethiopia

* tekakej@gmail.com

**Data Availability Statement:** All relevant data are within the manuscript and its Supporting information files.

## Abstract

Coffee, the second most traded commodity globally after petroleum and is the most exported cash crop of Ethiopia. However, coffee cultivation faces challenges due to fungal diseases, resulting in significant yield losses. The primary fungal diseases affecting coffee production include coffee berry disease, wilt disease (caused by *Gibberella xylarioides*), and coffee leaf rust. In this study, we aimed to isolate potentially antagonistic actinomycetes from the root rhizosphere of wild *Coffea arabica* plants in the Yayo coffee forest biosphere in southwestern Ethiopia. Soil samples were collected from the rhizosphere, and actinomycetes were selectively isolated and identified to the genus level by morphological, physiological, and biochemical characterization. These pure isolates were screened for their antagonistic activity against *Gibberella xylarioides in vitro* using a dual culturing method. Promising isolates demonstrating strong inhibition of fungal mycelial growth were further investigated through *in vivo* experiments using coffee seedlings. A total of 82 rhizobacteria were isolated. These isolates' inhibition of fungal mycelial growth varied from 0% to 83.3%. Among them, four isolates MUA26, MUA13, MUA52, and MUA14 demonstrated the highest percentage inhibition of fungal mycelial growth: 83.3%, 80%, 76.67%, and 73.3%, respectively. Seedlings inoculated with MUA13, MUA14, and MUA26 during the challenge inoculations (Rhizobacteria + *Gibberella xylarioides*) exhibited the lowest disease incidence compared to the infected fungi (P < 0.05). Notably, the seedlings inoculated with MUA26 demonstrated the highest disease control efficiency, reaching 83% (P < 0.05). MUA26 was found to produce extracellular enzymes, including chitinase, protease, and lipase, which acted as inhibitors. In summary, this study highlights that MUA26, among the actinomycete isolates, exhibited significant antagonistic activity against *Gibberella xylarioides* f.sp. coffea. Its efficacy in controlling coffee wilt disease, both *in vitro* and *in vivo*, positions it as a potential bioinoculant for managing coffee wilt disease.

## Introduction

Coffee is the world's second most traded commodity after petroleum and Ethiopia's most significant export. It contributes 27% of foreign exchange earnings and more than 25% of rural

**Funding:** The author(s) received no specific funding for this work.

**Competing interests:** The authors have declared that no competing interests exist.

and urban employment [1]. In Ethiopia, coffee is primarily grown by smallholder farmers for their livelihoods. More than a quarter of the country's population is directly or indirectly involved in coffee production, processing, trading, and marketing [2]. A significant portion of the coffee produced in Ethiopia is also consumed domestically, and the coffee ceremony is also one of the traditional Ethiopian cultures with great socializing value in the society, so coffee is strongly associated with Ethiopians in many ways.

Ethiopian coffee is of immense national and global importance because it is the country of origin of Arabica coffee and therefore has most of the Arabica gene pool and organic coffee production. Currently, coffee production in all coffee-growing areas of Ethiopia faces threats from various diseases, with fungal diseases being the most prominent. Among the fungal coffee diseases, coffee wilt disease (CWD) has been threatening coffee production in Ethiopia since 1958 [3, 4]. Coffee wilt disease (CWD) is a vascular wilt syndrome commonly referred to as tracheomycosis and is caused by *Gibberella xylarioides* (*Fusarium xylarioides*). The pathogen occurs on coffee trees in two stages of development: Gibberella as a sexual or perfect stage producing wind-borne ascospores, and *Fusarium* as an asexual or imperfect stage with splash-borne conidia. Infection usually occurs in the imperfect stage, which enters through wounds in the base of the stem. The fungus blocks the water supply in the vascular system and causes a typical brown discoloration. Once a tree is infected, there is no other option but to uproot the tree and burn it in place to reduce the likelihood of the infection spreading [5]. Therefore, coffee farmers called the disease "coffee AIDS" because the pathogen causes irreversible damage to the plant. CWD is known to affect all coffee species, including wild native lines in tropical Africa [6, 7]. It is one of the economically most important coffee diseases along with coffee berry disease caused by *Colletotrichum gloeosporioides* in Ethiopia. CWD is an emerged coffee disease due to gene alterations in *Fusarium xylarioides* sero-types. Nowadays, the prevalence and incidence of CWD are increasing significantly in coffee growing areas of the country. The average incidence of coffee wilt in Ethiopia ranges from 20–70%. In Ethiopia, coffee production (yield) at the farm level decreased by 37% due to CWD, resulting in a 67% decrease in income. Annual national crop losses attributable to CWD were 3360 tonnes, equivalent to US $3,750,976 in Ethiopia [8, 9].

Numerous attempts were made to control CWD in Ethiopia, including the use of resistant coffee varieties, environmental management, and the use of fungicides. However, the problem persisted, indicating the ineffectiveness of the methods used to control the pathogen. In addition, CWD is a soilborne pathogen, making the use of chemical treatments difficult. The emergence of fungicide-resistant pathogens, the high cost of fungicides, consumer preference for chemical-free products (organic coffee), and other negative environmental impacts have raised great concern and calls for another, safer additional method to control the disease. As an alternative, the use of biological control methods using phyto-beneficial microorganisms has been cited and advocated as one of the best strategies for integrated control of coffee wilt in the context of sustainable agriculture. Biological pest control using phytobeneficial microorganisms involves the reduction of disease-producing activities of a pathogen in its active or dormant state by one or more organisms, achieved naturally or by manipulation of the environment, host, or antagonists, or by mass introduction of one or more antagonists. Phytobeneficial microorganisms that produce secondary metabolites and fungi cell wall lytic enzymes are recognized as potential antagonists and are used as biocontrol agents. Some microorganisms occupy niches and prevent pathogen colonization, protecting plants from infection, while others induce the host immune system when colonizing roots. For the above reasons, biological control with phyto-beneficial microorganisms is considered an effective and acceptable alternative for soilborne pathogens, including *Fusarium xylaroides*, and is therefore considered the best alternative for CWD control.

There are several groups of soil microorganisms with antagonistic properties against plant pathogens [10]. Actinomycetes have great potential in controlling various pathogens as they produce antibiotics and other important secondary metabolites. Streptomyces species belong to actinomycetes and are important producers of enzymes such as chitinase, cellulases, peptidase, protease, xylanase, and other compounds that have great potential in inhibiting the growth of pathogens [11–13]. Previous reports indicated that the rhizobacterial isolates *Bacillus subtilis*, *T. viride*, and *T. harzianum* are potential antagonists of *Gibberella xylarioides*, but the biocontrol activity of actinomycete isolates against *Gibberella xylarioides* has not been investigated.

Therefore, the present study aimed to isolate potential actinomycetes that exhibit strong biocontrol activity against *Gibberella xylarioides* from rhizosphere of *Coffee arabica* plant grown naturally in Yayo forest, southwestern Ethiopia.

## Materials and methods

### Study site

Rhizospheric soil samples were collected from natural coffee grown in the Yayo coffee forest biosphere, which is about 564 km from Addis Ababa. The site was selected because coffee wilt disease is prevalent in the coffee forest system and accounts for 30%. It is located between 8° 0′ 0″ N, 35° 30′ 0″ E with an elevation of 1376–1890 meters above sea level. **Permission was obtained from the Yayo Woreda administrative office to access the sample collection site.** Laboratory experiments were conducted at the Mattu University microbiology laboratory, located in the main campus and 36 km from the sampling site. Mattu has a relatively cool tropical monsoon climate under the Köppen climate classification and is suitable for coffee cultivation. The mean annual maximum temperature is 30°C and the mean annual minimum temperature is 14°C. The maximum rainfall occurs in the three months of June to August and the minimum rainfall in December to January, and it is also evergreen almost all year round.

### Soil sampling and collection

Rhizospheric soil was collected aseptically. Briefly, the soil used for bacterial isolation was excavated from the 5–15 cm depth of roots of the coffee plants located at 15 locations in the Yayo coffee forest. Bulk soil was removed from plants by shaking vigorously by hand for 10 min before collecting the rhizospheric soil from the roots of coffee plants. From each site, 250 g of rhizospheric soil was collected in sterilized polyethylene bags. The samples were stored in an icebox at 4°C and transported to the applied microbiology laboratory of Mattu University for Analysis. All the samples were mixed to obtain the exact composition of the sample.

### Isolation of actinomycetes

Isolation and preservation of actinomycetes was carried out as previously reported [14, 15]. Briefly, soil samples from the rhizosphere (1 g) were suspended in normal saline (9 mL) and allowed to stand for 15 min. The suspension was serially diluted to final dilutions of $10^{-4}$, $10^{-5}$, and $10^{-6}$, and aliquots (0.1 ml) of each dilution were placed on Pridham's agar (1% glucose, 1% starch, 0.2% $(NH_4)_2SO_4$, 0.2% $CaCO_3$, 0.1% $K_2HPO_4$, 0.1% $MgSO_4$, 0.1% NaCl, and 1.2% agar) and water-proline agar (WA) (1% praline and 1.2% agar) supplemented with 25 μg/mL nalidixic acid and 50 μg/mL cycloheximide to prevent the growth of other bacteria and fungi, respectively. Plates were incubated at 28°C for 4–14 days. Isolated actinomycetes were further subcultured at 28°C and maintained on Seino's (1% starch, 0.3% N-Z amine type A, 0.1% yeast extract, 0.1% meat extract, 0.3% $CaCO_3$, and 1.2% agar) and Waksman agar slants (1% glucose,

0.5% peptone, 0.5% meat extract, 0.3% NaCl, and 1.2% agar) at 4˚C and -20 ˚C, respectively. Strains were subcultured to fresh media every 2 months. For long-term preservation, conidial or mycelial suspension in 25% glycerol was kept at -80 ˚C.

## Antagonistic study of isolates against *Gibberella xylarioides*

*In vitro* antagonism study was carried out as earlier described [16]. Briefly, for initial screening, a small fungal agar block (1 cm X 2 cm), from the leading margin (actively growing edge) of cultures propagated on potato dextrose agar for five to seven days at 25 ˚C was centrally placed on a pre-solidified nutrient medium. Exponentially grown (24 hours old) bacterial cultures (two isolates/plate) were streaked as broadband (making a straight short bar) approx. 3 cm away from mycellial block at two opposite edges of duplicate petri dishes (90 mm diameter). Plates were incubated at 25 ˚C for 7–10 days and potent rhizobacterial isolates were selected depending on their degree of inhibition.

The culture of *Gibberella xylarioides* was obtained from the Jimma Agricultural Research Centre (JARC). A plate inoculated with pathogenic fungi served as a control. The clear zone around the fungal growth was measured and the percentage of fungal radial growth inhibition (a clear zone between the edges of the fungal mycelia and the bacterial colonies) was calculated using the following formula.

$$\text{Inhibition (\%)} = \frac{C - T}{C} \times 100$$

Where,
T = fungal radial mycelial growth during dual culture (bacteria and fungus)
$C$ = fungal radial mycelial growth (without antagonistic bacteria)

The experiment was conducted in three replicates. Finally, the actinomycetes with the most potent inhibitory activity were selected for further experiments.

## Morphological and physiological characterization of selected isolates

Morphological characterizations of selected potential isolates were conducted following the general method of isolation and characterization of actinomycetes [14, 15]. Briefly, microscopic examination was performed using cellophane tape and cover slip-buried methods using a light microscope. Gram staining and lactophenol blue staining were performed to check the morphology of the cells, and other morphological features (Aerial mass color, substrate mycelium color, pigment production, spore chain morphology) were identified using the coverslip culture technique.

To determine physiological growth characteristics, isolates inoculated onto starch-casein agar plates were incubated at various temperatures (4, 9, 15, 20, 25, 30, 35, 40, and 45 ˚C) to determine the minimum, optimum, and maximum temperatures. To determine pH tolerance, isolates were inoculated onto starch-casein agar adjusted to pH 3, 5, 7, 9, and 11 using spreading techniques and incubated at 30 ˚C for 7 days and the result was recorded. To determine salt tolerance, isolates were inoculated on starch-casein agar with different NaCl concentrations (0, 3, 5, and 7%) and incubated at 28 ˚C for 7 days. The experiments were conducted in three replicates.

## Biochemical characterization of actinomycetes isolates

Biochemical characterization and genus identification of the potential isolates were performed according to Taddei *et al*. [17]. Catalase, oxidase, nitrate reduction, starch hydrolysis, casein hydrolysis, gelatin liquefaction, sugar fermentation, tyrosine, and xanthine tests were used for

genus confirmation. Methyl Red, MacConkey agar, and motility tests were performed according to standard microbiological procedures. Each experiment was conducted in three replicates.

## Lytic enzyme production

The ability of isolates to produce proteolytic enzymes was determined as earlier described [18]. Milk agar medium (skim milk powder- 100 g/L, peptone-5 g/L, and agar-15 g/L) was prepared and inoculated by the single striking of each isolate colony and the plates were incubated at 37 ˚C for 48 h. The formation of the clear zones around the isolated colonies indicates protease production. Celluse test was performed as earlier described elsewhere [19]. Briefly, carboxy-methylcellulose (CMC) agar medium (1.0% peptone, 1.0% carboxymethylcellulose (CMC), 0.2% $K_2HPO_4$, 1% agar, 0.03% $MgSO_4.7H_2O$, 0.25% $(NH_4)_2SO_4$ and 0.2% gelatin at pH 7) was prepared and inoculated by the single striking of each isolate colony and plates were incubated at 30 ˚C for 48 h. The incubated CMC agar plates were flooded with 1% Congo red and allowed to stand for 15 min at room temperature. Clear zones appeared around growing bacterial colonies indicating cellulose hydrolysis.

Lipase test was performed using Tributyrin agar medium. Briefly, Tributyrin agar medium (peptone 5 g, Yeast extract 15 g, agar 15 g, and Tributyrin 10 mL at pH 7.5 ±2) was prepared and inoculated by single striking of each isolate colony, and plates were incubated at 30±2 ˚C for 48 hours. The clear zone around the bacterial colonies shows the lipase enzyme's activity.

Chitinase test was performed using chitinase agar medium. Briefly, chitinase agar medium (colloidal chitin 1% (w/v), $Na_2HPO_4$ ($6$ g $L^{-1}$), NaCl ($0.5$ g $L^{-1}$), $KH_2PO_4$ ($3$ g $L^{-1}$); $NH_4Cl$ ($1$ g $L^{-1}$), yeast extract ($0.05$ g $L^{-1}$) and agar ($15$ g $L^{-1}$)) was prepared and inoculated by the single striking of each isolate colony and plates were incubated at 30±2 ˚C for 5 days. The clear zone around the bacterial colonies shows the activity of chitinase enzyme [20]. Each of the above experiments was conducted in three replicates.

## Phosphate and zinc solubilization test

The ability of rhizobacteria to solubilize phosphate was tested on Pikovskaya agar medium [21]. Briefly, to identify the phosphate-solubilizing bacteria, strains were streaked on Pikovskaya agar medium containing (per liter): 0.5 g yeast extract, 10 g dextrose, 5 g $Ca_3(PO_4)_2$, 0.5 g $(NH_4)_2SO_4$, 0.2 g KCl, 0.1 g $MgSO_4.7H_2O$, 0.0001 g $MnSO_4.H_2O$, 0.0001 g $FeSO_4.7H_2O$, and 15 g agar. After 3 days of incubation at 28 ˚C, the strains that formed a clear zone around the colonies were considered positive.

Zinc solubilizing activity was tested according to Mumtaz et al. [22]. Briefly, test organisms were inoculated into modified Pikovskaya medium (ingredients g $L^{-1}$), (glucose 10.0 g; ammonium sulfate 1.0 g; potassium chloride 0.2 g; dipotassium hydrogen phosphate 0.2 g; magnesium sulfate 0.1 g; yeast 0.2 g; distilled water 1000 mL, pH 7.0) containing 0.1% insoluble zinc compounds (ZnO, $ZnCO_3$, and ZnS) and incubated at 28 ºC for 48 hours, strains that formed a clear zone around the colonies were considered positive. Both phosphate and zinc solubilization tests were conducted in three replicates.

## Bioassay

**Coffee seed preparation.** The seeds of *C. arabica* (Catura rojo variety) were used for the bioassay. The seeds were obtained from Jimma Agricultural Research Center (JARC), Mattu branch. After removing the parchment, the seeds were surface sterilized with 1% sodium hypochlorite for 2 minutes. The surface sterilized seeds were rinsed several times with sterile

distilled water and soaked again in sterile distilled water for 24 hours after removing the parchment. The soaked seeds were used for the *in vitro* and *in vivo* experiments.

**Experimental design.** The experiment included only one factor, bacterial antagonistic isolates. Four rhizobacterial antagonist isolates (MUA13, MUA14, MUA52, and MUA26) were inoculated into coffee plants seven days before pathogen inoculation. The experiment was conducted in the greenhouse in the Randomized Complete Block Design (RCBD).

*In vivo experiment under green house.* Surface sterilized coffee seeds were soaked in distilled water for 48 hours. Soaked seeds (20 seeds per pot) were sown into heat-sterilized and moistened sandy soil in disinfected plastic pots. Water was applied every 2 days to maintain optimum moisture for seed germination, emergence, and growth of seedlings in the greenhouse.

Bacterial and fungal inoculums were prepared for seedling inoculation as earlier performed by Tiru *et al.* [23]. Briefly, the four rhizobacterial isolates showing better results during the *in vitro* experiments were used for the pot experiment when the seedlings reached the fully expanded cotyledon stage under greenhouse conditions. The stock culture of the potential rhizobacteria isolates was revived on actinomycetes isolation agar for 4 days. The colony-forming unit was counted using a colony counter, and then serial (1:10) dilution was made. Finally, CFU mL$^{-1}$ was calculated, and $10^9$ CFU mL$^{-1}$ bacterial cells were used for seedling inoculation. For the preparation of inoculum of pathogen, the stock culture of *G. xylaroids* was grown onto PDA for 7 days at 28+2 ˚C. The pure culture of *G. xylaroids* grown in plates for 7 days was flooded with 10 mL of sterile water and then rubbed gently from the agar surface to free the conidia. The spore suspension was uniformly stirred up with a sterilized magnetic stirrer and then filtered into another sterile beaker through a double layer of cheesecloth. Then spore suspension concentration was adjusted to 2.3 x $10^6$ conidial mL$^{-1}$.

When coffee seedlings reached the cotyledon stage (72 days after sowing), they were inoculated with a conidial suspension of *G. xylarioids* and rhizobacterial isolate by stem nicking procedure as earlier described by Tiru *et al.* [23]. A sterile scalpel was first immersed into the inoculum suspensions, and then the stem of each seedling was gently wounded with the scalpel at about 2 cm from the soil level. A drop (nearly 1 mL) of fungal and bacterial inoculums was placed in the notch at seven-day intervals. For the test of biocontrol activity, the bacterization of seedlings was made seven days before inoculation by the pathogen (*G. xylarioides*). The negative and positive controls were treated with sterile distilled water and the pathogen, respectively. To create favorable conditions for both pathogens and antagonists, chambers were made for 10 days in a greenhouse using a transparent thick polythene sheet. The temperature in the chambers was measured three times a day, and the mean temperature was between 22 ˚C and 26 ˚C. The relative humidity of the chamber was maintained above 85% by a humidifier. Then, the number of healthy and wilting seedlings was counted based on external symptoms for six months starting from 30 days after inoculation. The characteristics of external symptoms, number of healthy and infected coffee seedlings per pot were recorded at 14-day intervals for six months starting after month after inoculation with *G. xylarioides* and rhizobacterial isolates. The percentage of Disease Severity Index (DSI, %) and biological control efficiency were computed.

The percentage of disease incidence (DI, %) was calculated from the cumulative number of dead over the total number of seedlings (dead plus healthy).

$$DI\ (\%) = \frac{No.\ of\ infected\ seedlings}{Total\ No.\ of\ seedlings} \times 100$$

In addition, the disease severity was recorded with scales using 0 to 4 scales (classes). Where 0 = no disease, 1 = curling leaves and stunted growth, 2 = leaf wilting and yellowing, 3 = leaf

necrosis, leaf wilting and abscission and 4 = plants were dead. Disease Severity Index (DSI) for each treatment was expressed as a percentage of the maximum possible infection using the equation:

$$DSI\,(\%) = \frac{B + 2C + 3D + 4E}{4(A + B + C + D + E)} \times 100$$

Where, A is numbers of seedling in class 0, B is numbers of seedling in class 1, C is numbers of seedling in class 2, D is numbers of seedling in class 3 and E is numbers of seedling in class 4.

The biological control efficiency (BE, %) was calculated using below formula [24].

$$BE\,(\%) = \frac{\text{Disease index of control} - \text{Disease index of treatment}}{\text{Disease index of control}} \times 100$$

Where, disease index of control is the disease index from only pathogen inoculated seedlings, disease index of treatment is disease index from pathogen and biocontrol agent inoculated seedlings.

## Data analysis

The data were subjected to analysis of variance (one-way ANOVA) to compare different treatments using Statistical Package for the Social Sciences (IBM SPSS, Windows version 20) software. Tukey's test was performed at a 0.05 significance level to check for significant differences between treatments.

## Results

### Isolation of potential antagonistic actinomycetes and *in vitro* study

A total of 82 actinomycetes were selectively isolated from rhizospheric soil of natural *Coffee arabica* L. tree grown in Yayo coffee forest biosphere reserve, south western Ethiopia. The isolates were studied against potential coffee pathogen, *G. xylariodes* for radial mycelial growth inhibition by dual culture (on Potato Dextrose Agar + Yeast Extract medium). Four isolates (MUA13, MUA14, MUA52 and MUA26) showed greater than 70% mycelial growth inhibition of *G. xylariodies*. The isolate designated as MUA26 showed the highest *G. xylariodies* mycelial growth inhibition, which was 83% (Fig 1 and S1 Table).

### Morphological characterization of potential isolates

The four potential isolates developed well-grown colonies on Seino's agar after 7 days of incubation at 28°C. The colonies of all isolates were rounded in shape. The aerial mycelia of isolates MUA13, MUA14, and MUA52 were white, while that of MUA26 was yellow. The substrate mycelia of isolates MUA13, MUA14, and MUA26 were white, while that of MUA52 was light yellow. MUA26 produced a yellow water-soluble pigment while the others did not (Table 1). The isolates possessed a powdery texture. The isolates were gram-positive, aerobic, and filamentous except for MUA14 which was spherical, possessed branched filaments and produced oval spores arranged in short to long chains or singly. The four isolates are non-motile. Based on these characteristics, the isolates were primarily identified as *Streptomyces* strains (abundant aerial mycelium with powdery spores). Other physiological characteristics of the isolates such as pH, NaCl, and temperature tolerance of potent isolates were summarized in Table 1 and S2 Table.

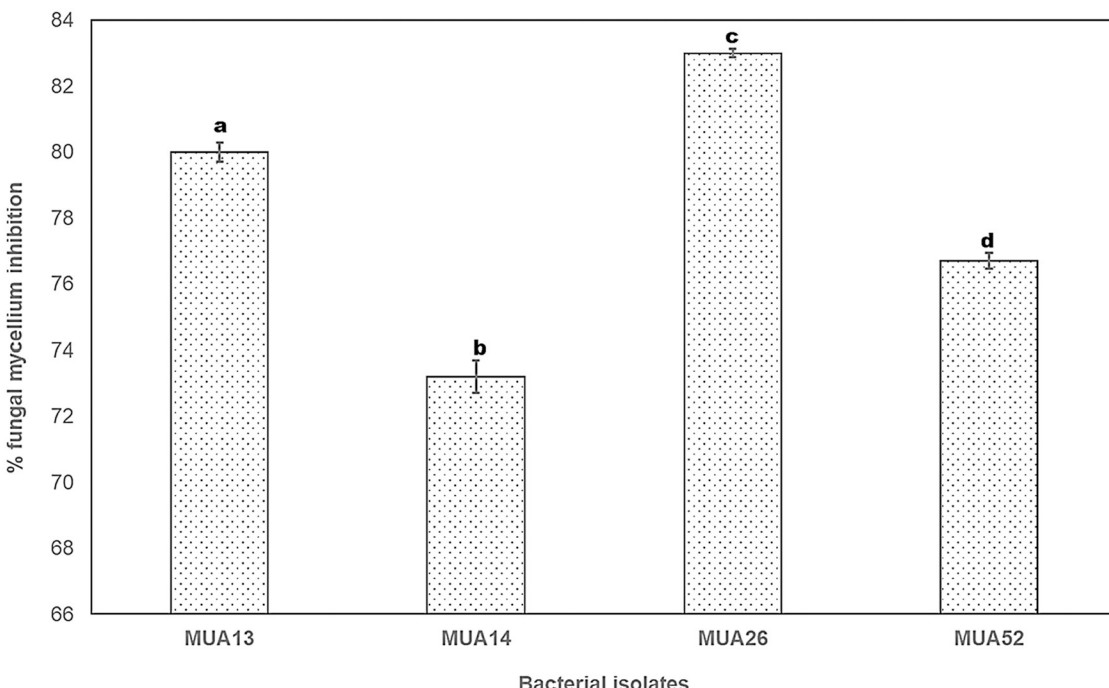

**Fig 1. *In vitro*, inhibition of *G. xylarioides* mycelial growth by rhizobacteria isolated from the rhizosphere of *Coffea arabica* L.** Each value is the average of three duplicate samples with standard error, and various alphabets shown in superscript indicate mean treatments that are substantially different according to Tukey's HSD posthoc test at $p \leq 0.05$.

## Biochemical characteristics, lytic enzyme production, and mineral solubilization of the isolates

The biochemical characteristics, lytic enzyme production, and mineral solubilization of the isolates were summarized, recorded, and presented in Table 2. All isolates were positive for catalase, oxidase, caseine, xanthine, tyrosine and gelatin hydrolysis but negative for the Mac-Conkey test. Except for MUA26, all isolates were methyl red test negative. All the isolates showed positive results for the utilization of D-Glucose, D-Fructose, Starch, Lactose, and Dextrose as a carbon source and Yeast extract as a nitrogen source (Table 2).

**Table 1. Morphology and physiological growth characters of actinomycete isolates.**

| Isolates | Description (Macroscopic and Microscopic, and growth characteristics) |
|---|---|
| MUA13 | Aerial mycelium white, moist, circular, colony reverse off white, Filaments branched, spores oval in short chains.<br>Temp. range 10–45 ˚C (opt. 30 ˚C), pH range 2–11 (opt. 7), salt tolerance up to 7% |
| MUA14 | Aerial mycelium white, smooth, circular, colony reverse off white, Filaments branched, spores oval in long chains.<br>Temp. range 10–45 ˚C (opt. 30 ˚C), pH 3–11 (opt. 7), salt tolerance up to 7% |
| MUA26 | Aerial mycelium yellow, rough, circular, colony reverse off white<br>Filaments branched, spores oval in short chains.<br>Temp. range 10–45 ˚C (opt. 30 ˚C), pH 3–11 (opt. 7), salt tolerance up to 7% |
| MUA52 | Aerial mycelium white, smooth, circular, colony reverse yellowish Filaments branched, spores oval in long chains<br>Temp. range 10–45 ˚C (opt. 30 ˚C), pH 4–11 (opt. 7), salt tolerance up to 5% |

Opt Optimum, Temp temperature

**Table 2. Biochemical characteristics, lytic enzyme production, and mineral solubilization of actinomycete isolates.**

| Biochemical characteristics | Isolate no | | | |
|---|---|---|---|---|
| | MUA13 | MUA14 | MUA26 | MUA52 |
| Oxidase | + | + | + | + |
| Catalase | + | + | + | + |
| Gelatinase | + | + | + | + |
| Urease | + | - | + | + |
| Methyl Red | - | - | + | - |
| MacConkey test | - | - | - | - |
| Utilization of carbon and nitrogen sources | | | | |
| D-Glucose | + | + | + | + |
| D-Fructose | + | + | + | + |
| Starch | + | + | + | + |
| Lactose | + | + | + | + |
| Dextrose | + | + | + | + |
| Caseine | + | + | + | + |
| Xanthine | + | + | + | + |
| Tyrosine | + | + | + | + |
| Yeast extract | + | + | + | + |
| Production of lytic enzymes | | | | |
| Protease | + | + | + | + |
| Lipase | - | - | + | + |
| Cellulase | - | - | - | - |
| Chitinase | + | + | + | + |
| Mineral solubilization | | | | |
| Phosphate solubilization | + | + | + | + |
| Zinc solubilization | + | + | + | + |

Note;—negative (absent), + positive (present)

The rhizobacteria isolates varied in terms of the production of hydrolytic enzymes, viz. protease, lipase, cellulase, and chitinase. All isolates were protease-positive and cellulase-negative. Isolate MUA13 and MUA14 were lipase negative, whereas isolate MUA26 and MUA52 were positive. All the isolates (MUA13, MUA14, MUA26, and MUA52) efficiently solubilized Zinc oxide. Except for MUA14, all solubilized tricalcium phosphate. In addition, all the isolate colonies possessed an earthy odor.

### *In vivo* antagonistic study

The inoculation of *C. arabica* seedlings by the rhizobacterial antagonists 7 days before the pathogen *G. xylarioides* significantly reduced disease incidence and severity. The treatment of *C. arabica* seedlings with MUA13, MUA14, and MUA26 reduced the incidence of coffee wilt disease to 33.3% compared to fungi infected (Table 3 and S3 Table). There was no significant variation in CWD incidence among the three isolates (MUA13, MUA14, and MUA26). The highest reduction of coffee wilt severity was recorded in Coffee arabica seedlings treated with MUA26, which was reduced from 100% on positive control (infected by *G. xylarioides* only) to 16.7% (Table 3 and S4 Table). The least reduction of coffee wilt incidence and severity was recorded in coffee arabica seedlings inoculated by MUA52, which was 66.7%. There was a significant variation in coffee wilt disease severity between the four rhizobacterial antagonists

**Table 3. The percentage of coffee seedling death and disease incidence after inoculation of *G. xylarioides* and potential bacterial isolates.**

| Treatment | Disease severity (%) | Disease incidence (%) | Disease control efficiency (%) |
|---|---|---|---|
| MUA13+*G.xylarioides* | 33.3±4.8[a] | 33.3±5.6[a] | 66.7±4.8[a] |
| MUA14+*G.xylarioides* | 25.0±6.3[b] | 33.3±6.9[a] | 75±6.3[b] |
| MUA26+*G.xylarioides* | 16.7±3.7[c] | 33.3±6.6[a] | 83.3±3.7[c] |
| MUA52+*G.xylarioides* | 50.0±6.1[d] | 66.7±7.2[b] | 50 ±6.1[d] |
| Control (N) | 00.0±0.0[e] | 0.0±0.0[c] | - |
| Control (P) | 100±0.0[f] | 100±0.0[d] | 0[e] |

Means with different letters are significantly different across columns at p≤0.05 according to the Tukeys test. Control (N): negative control (only sterile distilled water inoculated seedlings); Control (P): positive control (only *G. xylarioides* infected).

(p<0.05). The highest percentage of biological control was obtained against *G. xylarioides* using MUA26, followed by MUA14 and MUA13, which were 83.3%, 75%, and 66.7% respectively (Table 3 and S5 Table). There was a significant variation in disease control efficiency among the rhizobacterial isolates (p<0.05). The least biocontrol efficiency was recorded when MUA52 was inoculated to coffee arabica seedlings seven days before the pathogen (Table 3).

## Discussions

In numerous countries, chemical fertilizers and pesticides have long been obligatory agricultural inputs to enhance crop production and productivity. However, in recent times, due to their high costs and environmental safety concerns, many countries worldwide have been advocating bioinoculants as an alternative to chemical fertilizers and pesticides [25–28]. These bioinoculants consist of phytobeneficial bacteria associated with plant rhizospheres, which promote plant growth through direct and indirect mechanisms. They are recognized as a safe and superior alternative to chemical-based fertilizers and pesticides [29, 30]. Consequently, there is a growing demand to employ them as bioinoculants in organic farming. Researchers globally have been searching for potential isolates from diverse ecological regions for this purpose [31–35].

The current study focuses on isolating bacteria from the genus actinomycetes that can antagonize the growth of pathogenic fungi (specifically, *Gibberella xylarioides*) responsible for coffee wilt disease. This disease significantly impacts coffee production and is challenging to control using chemical pesticides [5, 36]. We have chosen actinomycetes deliberately for the current study because of their ability to produce bioactive compounds and other survival-enhancing characteristics in soil. These features have garnered interest in sustainable agriculture as sources of biologically active compounds, biocontrol agents, and plant growth-promoting rhizobacteria (PGPR), making them ideal for commercial microbial applications [13, 14, 37]. Streptomyces, the most common actinomycete found in the environment, constitutes the primary group with significant agricultural applications as highlighted by Silva *et al.* [38].

In a recent study, we selectively isolated a total of 82 actinomycetes. Among these, four specific isolates MUA13, MUA14, MUA26, and MUA52 demonstrated potent inhibition of *Gibberella xylarioides* mycelium growth, with a zone of inhibition exceeding 70%. Remarkably, MUA26 exhibited the highest percentage of fungal radial inhibition (83.3%), while MUA14 showed the lowest (73.3%). This is the first report on the antagonism of actinomycetes against *G. xylarioides* hence the results can be used as a reference for future studies. Earlier reports indicated the mycelial growth inhibition profiles of the *Trichoderma* isolates against *F.*

*xylarioides* ranged from 44.5% to 84.8% [6]. Tiru *et al.* (2013) reported Bacillus spp (JU544) isolated from *Coffea arabica* L. showed the highest percentage of *G. xylarioides* mycelial growth inhibition (71.5%) [23]. Numerous studies suggest that the rhizosphere is ideal for isolating actinomycetes with inhibitory activity against plant fungal pathogens [11, 12, 14, 38]. The variation in fungal mycelial inhibition of the isolates may be attributed to differences in the isolates' capacity to release various lytic enzymes and metabolites against the pathogen.The isolates colonies were, slow-growing, round, gram-positive, aerobic, powdery, folded, and with aerial and substrate mycelia of different colors. This indicates the isolates are morphologically similar to the actinomycetes group which was also shown in Bergey's Manual of Determinative Bacteriology [15]. Streptomyces colonies, characterized by their opaque, rough, nonspreading morphology, are typically embedded in the agar medium [14]. In addition, all colonies produced an earthy odor, a peculiar feature of actinomycetes. All the isolates could also degrade casein, tyrosine, and xanthine, which confirms the genus-level classification of the isolates as Streptomyces and this result was similar to earlier studies on morphological and biochemical characterization of actinomycetes [14, 17]. Therefore, based on morphological (macroscopic and microscopic) and biochemical characterization of the four isolates, the isolates were classified as members of the genus actinomycetes, particularly as members of streptomyces species. These findings align with previous reports by Shirling & Gottlieb and Taddei *et al.* [15, 17].

Under greenhouse conditions, the effectiveness of four Streptomyces strains in mitigating coffee wilt disease was assessed. All tested strains significantly reduced the disease severity caused by artificial inoculation of the coffee pathogen *Gibberella xylarioides* in coffee seedlings, compared to the control group. Among these strains, MUA26 exhibited the lowest coffee wilt disease severity (16.7%) and the highest biocontrol efficiency (83.3%). This suggests that MUA26 holds promise as a biocontrol agent against the coffee wilt disease pathogen. The superior biocontrol efficiency of MUA26 may be attributed to its production of extracellular lytic enzymes that directly impact the cell wall of pathogenic fungi synergistically. Specifically, MUA26 produces chitinase, protease, and lipase, whereas the other three strains produce one or two of these enzymes. These lytic enzymes play a crucial role in hydrolyzing the cell walls of pathogenic fungi, indirectly promoting plant growth by suppressing fungal proliferation [13, 14, 39]. Thus, these isolates' production of lytic enzymes contributes to inhibiting *G. xylarioides* growth both *in vitro* and *in vivo*. In addition to their biocontrol activity, the four isolates (MUA13, MUA14, MUA26, and MUA52) also solubilized phosphate and zinc from their insoluble inorganic forms. The solubilization of P and Zn benefits plants by making these essential nutrients more available. Adequate phosphorus (P) is crucial for optimal crop production and health from the early growth stages. However, limited P supply often hampers crop yield. While P fertilizers are commonly applied to ensure sufficient availability, only a small portion of this P is immediately accessible for crop uptake, with the remainder converting into insoluble complexes. Phosphate-solubilizing rhizobacteria are crucial in mobilizing insoluble inorganic phosphates from their complex forms to the bulk soil, where plant roots can access them. Phosphorus is a vital nutrient for sustainable plant productivity [40]. Similarly, zinc, an essential micronutrient, plays a critical role in plant growth by participating in enzymatic reactions, metabolic processes, and redox reactions [41, 42]. When zinc is supplied to plants as fertilizers (e.g., zinc sulfate), it transforms into various insoluble complexes based on soil types and chemical reactions. The ability of these isolates to solubilize both phosphate and zinc from their insoluble forms enhances positive interactions between plants and microbes, ultimately leading to bacterial colonization of plant roots. Thus, the isolates can stay longer in the rhizosphere and have long-term effects in protecting the coffee plant from the pathogen.

## Conclusions

The results of our study demonstrate that the four actinomycetes strains function as plant growth-promoting rhizobacteria under both in vitro and in vivo conditions. Notably, the isolate MUA26 exhibited promising outcomes in both experimental settings, suggesting its potential as a biocontrol agent against *G. xylarioides*. Furthermore, the phosphorus and zinc solubilization abilities of these isolates benefit plant health and foster positive plant-microbe interactions. Therefore, the selected strains, particularly MUA26, hold promise as biocontrol agents against *G. xylarioides*.

## Supporting information

**S1 Table.** *In vitro*, **inhibition of** *Gibberella xylarioides* **mycelial growth by rhizobacteria isolates.**
(DOCX)

**S2 Table. The growth of selected rhizobacterial isolates under different pH, temperature, and % of NaCl.** +++: abundant growth, ++: medium growth, +: slow growth, -: No growth.
(DOCX)

**S3 Table. Effect of inoculation of rhizobacteria isolates on reduction of coffee wilt disease incidence caused by** *G. xylarioides* **under greenhouse conditions.** r = replicate; Control (N): negative control (only sterile distilled water inoculated seedlings); Control (P): positive control (only *G. xylarioides* infected).
(DOCX)

**S4 Table. Effect of inoculation of rhizobacteria isolates on reduction of coffee wilt disease severity caused by** *G. xylarioides* **under greenhouse conditions.** r = replicate; Control (N): negative control (only sterile distilled water inoculated seedlings); Control (P): positive control (only *G. xylarioides* infected).
(DOCX)

**S5 Table. The biological control efficiency (BE, %) of rhizobacterial isolates against coffee wilt disease caused by** *G. xylarioides* **under greenhouse conditions.** r = replicate; Control (N): negative control (only sterile distilled water inoculated seedlings), Control (P): positive control (only *G. xylarioides* infected).
(DOCX)

## Author Contributions

**Conceptualization:** Mimi Nuguse, Tekalign Kejela.

**Data curation:** Mimi Nuguse.

**Formal analysis:** Mimi Nuguse.

**Investigation:** Mimi Nuguse.

**Methodology:** Mimi Nuguse, Tekalign Kejela.

**Resources:** Mimi Nuguse.

**Software:** Mimi Nuguse, Tekalign Kejela.

**Supervision:** Tekalign Kejela.

**Validation:** Mimi Nuguse.

**Visualization:** Mimi Nuguse.

**Writing – original draft:** Mimi Nuguse, Tekalign Kejela.

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
