## [Decision Letter · Decision Letter 0]

7 May 2024

PONE-D-24-10862Actinomycetes Isolated from Rhizosphere of Wild Coffea arabica L. Showed Strong Biocontrol Activities against Coffee Wilt DiseasePLOS ONE

Dear Dr. Geleta,

Thank you for submitting your manuscript to PLOS ONE. After careful consideration, we feel that it has merit but does not fully meet PLOS ONE’s publication criteria as it currently stands. Therefore, we invite you to submit a revised version of the manuscript that addresses the points raised during the review process.

We look forward to receiving your revised manuscript.

Kind regards,

Ali Tan Kee Zuan, Ph.D.

Academic Editor

PLOS ONE

Journal Requirements:

Reviewers' comments:

Reviewer's Responses to Questions

**Comments to the Author**

1. Is the manuscript technically sound, and do the data support the conclusions?

Reviewer #1: No

Reviewer #2: Partly

2. Has the statistical analysis been performed appropriately and rigorously? 

Reviewer #1: Yes

Reviewer #2: Yes

3. Have the authors made all data underlying the findings in their manuscript fully available?

Reviewer #1: No

Reviewer #2: Yes

4. Is the manuscript presented in an intelligible fashion and written in standard English?

Reviewer #1: Yes

Reviewer #2: No

5. Review Comments to the Author

Reviewer #1: The authors' diligent work has revealed the potential of Actinomycetes isolates as biocontrol agents for Coffee Wilt Disease (CWD). These findings are significant for the scientific community and hold great promise for implementing sustainable agricultural practices in the coffee industry.

This manuscript does not clearly state the number of replications prepared for some tests conducted. In addition, data points behind the mean and variance measures of the results presented were unavailable in the manuscript. It needs improvement.

The isolates were not deposited in any Microbial culture collection center - no information was provided in the manuscript.

Additional results (if available) on molecular identification, phylogenetic analysis of the isolates and electron micrographs (SEM and TEM) of cell walls of treated plants will make the manuscript more interesting for publication in PLOS ONE.

Reviewer #2: General comments:

1. This study is interesting and demanding for the sustainability of coffee industry.

2. English need to be improved.

3. Found many inconsistencies in words, spelling, formatting etc.

4. Lack of discussion on the findings.

5. No work done on molecular identification or any other appropriate identification on the selected actinomycetes isolates, therefore this manuscript should be rejected.

Specific comments:

Line 49, important export?

Line 97, typo Fusarium

Line 107, delete ‘root rhizosphere’, delete root

Line 114, 1376–1890 mas, mas?

Line 116, Mattu is located in the temperate zone

Line 117, ideal for arabica coffee plantation

Line 125-126, describe in detail how field samples can be collected aseptically?

Line 147, fungal agar block, check 1 x 2 cm2? Wrong?

Line 147 , what do you mean by leading margin of cultures, please explain.

Line 148, check degree symbol, 25 °C

Line 153, delete ‘free of charge’, was obtained from

Line 162, delete rhizospheric, to actinomycetes isolates with the most.....

Line 181, delete ‘test’ after Catalase, replace with tests after xanthine

Line 183, delete test after methyl red and MacConkey. Typo ‘Methy’

Line 184, delete ‘Cliques’

Line 189, ‘clean? zones’, to ‘clear zones’

Line 199, delete ‘3’ at end line

Line 241, small letter ‘a’ for ‘actinomycetes’ not capital Actinomycetes, and more throughout this manuscript, please check, unless use as starting word in a sentence.

Line 269, typo, ‘seedlings’

Line 290, small letter ‘a’ for actinomycetes, delete ‘root’, rhizospheric soil

6. PLOS authors have the option to publish the peer review history of their article (what does this mean?). If published, this will include your full peer review and any attached files.

Reviewer #1: No

Reviewer #2: No

---

## [Author Response · Author response to Decision Letter 0]

20 Jun 2024

Date 20/06/2024

To 

The Editorial manager,

PLOS ONE

Subject: submission of revised manuscript for consideration of its publication in PLOS ONE

Dear Sir/Madam,

I am writing to submit our revised manuscript entitled, “Actinomycetes isolated from rhizosphere of wild Coffea arabica L. showed strong biocontrol activities against coffee wilt disease,” for consideration of its publication in the esteemed journal PLOS ONE. First of all, I would like to thank you for allowing us to revise and submit the manuscript. I appreciate the time and details the academic editor and reviewers provided to enrich our manuscript. Based on the comments, we have thoroughly revised the whole manuscript. As commented by the academic editor, the manuscript was edited based on PLOS ONE's style requirements. The full name of the authority that approved the field site access was included in the material method section under the sub-topic ‘study site’. All relevant data are within the manuscript and the details were also added as supporting information in a separate file. We hope you will be pleased with this revision and consider it for publication; we couldn't have done it without your input. Responds to each point raised by the reviewers are indicated below. 

Sincerely,

Tekalign Kejela (Ph.D)

Author response to reviewers' Comments

Reviewer #1: 

 The authors' diligent work has revealed the potential of Actinomycetes isolates as biocontrol agents for Coffee Wilt Disease (CWD). These findings are significant for the scientific community and hold great promise for implementing sustainable agricultural practices in the coffee industry.

 Authors' response: we appreciate the reviewer's commitment to critically reviewing our manuscript. We agree with the reviewer’s idea and thus why we conducted the study.

 This manuscript does not clearly state the number of replications prepared for some tests conducted. In addition, data points behind the mean and variance measures of the results presented were unavailable in the manuscript. It needs improvement.

 Authors' response: All the experiments were conducted in three replicates and this information was added as a footnote under each figure and table in the manuscript. In addition, data points behind the mean and variance were incorporated as necessary and uploaded as a separate file as suggested by the reviewer. 

 The isolates were not deposited in any Microbial culture collection center - no information was provided in the manuscript.

 Authors' response: the isolate was deposited in the Applied microbiology laboratory of Mattu University, Mettu, Ethiopia. This information was incorporated in the MS as suggested by the reviewer. NB: No microbial culture collection center established so far in Ethiopia.

 Additional results (if available) on molecular identification, phylogenetic analysis of the isolates and electron micrographs (SEM and TEM) of cell walls of treated plants will make the manuscript more interesting for publication in PLOS ONE.

 Authors' response: we appreciate the reviewer’s comment, we performed the identification process to genus level based on morphological and biochemical characteristics. We don’t have the mentioned data to include in the MS. We performed the experiments to the best of our resources. We will consider it in our future work. 

Reviewer #2

General comments:

 This study is interesting and demanding for the sustainability of coffee industry.

 Authors' response: we appreciate the reviewer’s commitment to critically reviewing our manuscript. We agree with the reviewer’s idea and thus why we conducted the study. Using bioinoculants in the coffee industry is tremendously important due to emerging pesticide-resistant pathogens, and the difficulty of controlling soil-born pathogens including G. xylarioides using chemical pesticides as it affects non-targeted populations. Furthermore, the customer performance of chemical-free products and environmental safety concerns make biocontrol a demanding input in the sustainable coffee industry. 

 English need to be improved.

 Authors' response: grammatical mistakes, spelling errors, punctuations, and other language issues checked throughout the manuscript and corrected.

 Found many inconsistencies in words, spelling, formatting etc.

 Authors' response: inconsistencies in words, spelling, and formatting throughout the manuscript checked and corrected 

 Lack of discussion on the findings.

 Authors' response: The discussion on the findings was incorporated, and corrected as per the suggestion. 

 No work done on molecular identification or any other appropriate identification on the selected actinomycetes isolates, therefore this manuscript should be rejected.

 Authors' response: we did not perform molecular identification of the isolates, but, we have performed the appropriate identification method of the isolates to genus level to the best of our resources as outlined below

 We used the selective isolation method of actinomycetes 

 Primary identification of the isolates to the genus level was performed by morphological (macroscopic and microscopic) and biochemical characterization of the isolates based on Bergey’s Manual of Determinative Bacteriology. 

 In addition, for the confirmation of genus-level taxonomic identification of the isolates, gram reaction, caisin, xanthine, and tyrosine tests were performed according to Taddie et al, 2006. 

Hence, we used an appropriate and scientifically acceptable method to classify the rhizobacteria isolates to genus level, and thus why we reported actinomycete isolates. We appreciate the reviewer's comment and will consider species-level identification using molecular methods in our future work. 

Specific comments:

Line 49, important export?

 Authors' response: ‘important export’ corrected to ‘significant export’

Line 97, typo Fusarium

 Authors' response: ‘Fussarium’ corrected to ‘Fusarium’

Line 107, delete ‘root rhizosphere’, delete root

 Authors' response: ‘root rhizosphere’ corrected to ‘rhizosphere’ as per suggestion 

Line 114, 1376–1890 mas, mas?

 Authors' response: ‘1376–1890 mas’ corrected to ‘1376–1890 meters above sea level’

Line 116, Mattu is located in the temperate zone

 Authors' response: ‘Mattu is located in the temperate zone’ corrected to ‘Mattu has a relatively cool tropical monsoon climate under the Köppen climate classification’

Line 117, ideal for arabica coffee plantation

 Authors' response: ‘ideal for arabica coffee plantation’ corrected to “which is suitable for coffee cultivation”

Line 125-126, describe in detail how field samples can be collected aseptically?

 Authors' response: details of how field samples collected were incorporated 

Line 147, fungal agar block, check 1 x 2 cm2? Wrong?

 Authors' response: ‘fungal agar block (1 x 2 cm2 )’ corrected to ‘fungal agar block (1 cm X 2 cm)’

Line 147 , what do you mean by leading margin of cultures, please explain.

 Authors' response: The leading margin of cultures refers to the actively growing edge of the culture. Explanation incorporated as per suggestion.

Line 148, check degree symbol, 25 °C

 Authors' response: the incorrect symbol ‘ 0C’ corrected to ‘°C’

Line 153, delete ‘free of charge’, was obtained from

 Authors' response: the word was deleted

Line 162, delete rhizospheric, to actinomycetes isolates with the most.....

 Authors' response: the word was deleted

Line 181, delete ‘test’ after Catalase, replace with tests after xanthine

 Authors' response: corrected as per suggestion 

Line 183, delete test after methyl red and MacConkey. Typo ‘Methy’

 Authors' response: corrected as per suggestion, ‘Methy’ corrected to ‘methyl” 

Line 184, delete ‘Cliques’

 Authors' response: the word was deleted

Line 189, ‘clean? zones’, to ‘clear zones’

 Authors' response: ‘clean zones’ corrected to ‘clear zones’ 

Line 199, delete ‘3’ at end line

 Authors' response: the number ‘3’ was deleted

Line 241, small letter ‘a’ for ‘actinomycetes’ not capital Actinomycetes, and more throughout this manuscript, please check, unless use as starting word in a sentence.

 Authors' response: throughout the manuscript ‘Actinomycetes ‘is corrected to ‘actinomycetes’ except when used as a starting word in a sentence.

Line 269, typo, ‘seedlings’

 Authors' response: The incorrect word ‘seedlidgs’ in the formula DI (% )=(No.of infected seedlidgs)/(Total No.of seedlings)×100 corrected to ‘seedlings’ and the formula corrected to DI (%)=(No.of infected seedlings)/(Total No.of seedlings)×100

Line 290, small letter ‘a’ for actinomycetes, delete ‘root’, rhizospheric soil

 Authors' response: ‘Actinomycetes ‘ corrected to ‘actinomycetes’ and the word ‘root’ deleted

---

## [Editor Report · Decision Letter 1]

25 Jun 2024

Actinomycetes isolated from rhizosphere of wild Coffea arabica L. showed strong biocontrol activities against coffee wilt disease

PONE-D-24-10862R1

Dear Dr. Geleta,

We’re pleased to inform you that your manuscript has been judged scientifically suitable for publication and will be formally accepted for publication once it meets all outstanding technical requirements.

Kind regards,

Ali Tan Kee Zuan, Ph.D.

Academic Editor

PLOS ONE
---

## [Editor Report · Acceptance letter]

28 Jun 2024

PONE-D-24-10862R1 

PLOS ONE

Dear Dr. Kejela, 

I'm pleased to inform you that your manuscript has been deemed suitable for publication in PLOS ONE. Congratulations! Your manuscript is now being handed over to our production team.

Kind regards, 

on behalf of

Dr. Ali Tan Kee Zuan 

Academic Editor

PLOS ONE